# The Prevalence and Impact of Polycystic Ovary Syndrome in Recurrent Miscarriage: A Retrospective Cohort Study and Meta-Analysis

**DOI:** 10.3390/jcm9092700

**Published:** 2020-08-21

**Authors:** Daniel Mayrhofer, Marlene Hager, Katharina Walch, Stefan Ghobrial, Nina Rogenhofer, Rodrig Marculescu, Rudolf Seemann, Johannes Ott

**Affiliations:** 1Clinical Division of Gynecologic Endocrinology and Reproductive Medicine, Department of Obstetrics and Gynecology, Medical University of Vienna, Spitalgasse 23, 1090 Vienna, Austria; daniel.mayrhofer@gmail.com (D.M.); marlene.hager@meduniwien.ac.at (M.H.); katharina.walch@meduniwien.ac.at (K.W.); stefan.ghobrial@meduniwien.ac.at (S.G.); 2Division of Gynecological Endocrinology and Reproductive Medicine, Department of Obstetrics and Gynecology, University Hospital LMU Munich, Marchioninistraße 15, 81377 Munich, Germany; Nina.Rogenhofer@med.uni-muenchen.de; 3Department of Laboratory Medicine, Medical University of Vienna, Spitalgasse 23, 1090 Vienna, Austria; rodrig.marculescu@meduniwien.ac.at; 4Department of Oral and Maxillofacial Surgery, Medical University of Vienna, Spitalgasse 23, 1090 Vienna, Austria; rudolf.seemann@gmail.com

**Keywords:** polycystic ovary syndrome, polycystic ovarian morphology, recurrent miscarriage, prevalence, risk factors, meta-analysis

## Abstract

*Background*: The use of different definitions and diagnostic approaches of polycystic ovary syndrome (PCOS) and recurrent miscarriage (RM) has led to a wide range of prevalence rates in the literature. Despite the persistent controversy about the factual prevalence of PCOS in RM, a vast number of studies have revealed evidence about their association with each other. The goals of this study were to evaluate the prevalence of polycystic ovarian morphology and PCOS within the RM population, performing meta-analyses with the obtained data from this study, together with previous reports on this topic and evaluating reproductive outcome in women with RM and PCOS. *Methods*: A retrospective cohort study with 452 women with RM and a meta-analysis were conducted. The main outcome parameter was the prevalence of PCOS in RM patients. *Results*: In the retrospective study, the prevalence of PCOS in RM was 9.5%. Negative results for the selected risk factors for RM were present in 283 patients (62.6%). From all evaluated possible underlying causes for RM, only the presence of thrombophilic disorders was significantly associated with PCOS (PCOS: 20.9% versus no PCOS: 7.8%, *p =* 0.010). In the meta-analysis of three studies on PCOS in RM patients, which used the revised Rotterdam criteria for defining PCOS, an estimated pooled prevalence of 14.3% (95% CI: 6.2–24.9) was found. In the retrospective data set, women in the PCOS group revealed significantly higher luteinizing hormone (LH), testosterone, and Anti-Mullerian hormone (AMH) levels than age- and body mass index (BMI)-matched controls with RM negative for the selected risk facotrs (*p <* 0.05). The rate of further miscarriages was significantly higher in PCOS women than in controls (71.4% versus 53.6%, respectively; *p =* 0.031). *Conclusions*: The prevalence of PCOS seems slightly increased in women with RM. Women with PCOS suffering from RM showed a significantly higher risk for further miscarriage and decreased chances of having a life birth of about 18% which did not reach statistical significance. Therefore, we assume that PCOS plays a moderate role in RM.

## 1. Introduction

Polycystic ovarian syndrome (PCOS) is a clinically and biochemically heterogeneous disorder and among the most frequent endocrine disorders in the field of gynecology [1,2]. Depending on the criteria used, its prevalence ranges from 6.5–19.9% of women of reproductive age [1,2]. Characteristic symptoms are menstrual dysfunction, sonographic evidence of polycystic ovaries, elevated serum androgens and insulin resistance. Other frequent features include infertility, acne, hirsutism, obesity, and alopecia [1]. These various and variable features have led to misconceptions regarding the definition of the syndrome in the past. Nowadays, the most widely used diagnostic criteria are the “Revised Rotterdam Criteria” [1,3]. Noteworthy, PCOS seems to have a substantial negative effect on fertility. Due to disturbances in follicular maturation, follicle growth often stops at a follicle size of 4–8 mm which prevents the development of a dominant follicle and, therefore, ovulation. In addition, once a pregnancy is achieved, spontaneous miscarriages can also be observed more frequently in PCOS patients; the risk has been reported to range from 42 to 73% [2].

Miscarriage is among the most common complications during pregnancy and can be sporadic or recurrent [4]. According to the World Health Organization (WHO), recurrent miscarriage (RM) is defined as three or more consecutive pregnancy losses before the 20th gestation week [5,6]. Primary RM is defined as having no previous live births, while patients with secondary RM already had at least one live birth [7,8]. It is estimated that five percent of all couples trying to conceive are affected by two consecutive miscarriages and that one percent is affected by three or more [4,7,8]. In 50% of the cases of RM, the underlying cause remains unknown [4,8,9]. Potential pathologies leading to RM include genetic abnormalities, infection, immune dysfunction, endocrine disorders, antiphospholipid syndrome, thrombophilic disorders, uterine pathologies, and cervical weakness [4,10].

In recent years, numerous studies have investigated the association between PCOS and RM. Increased rates of PCOS have been reported for women with RM [11]. However, the actual prevalence is controversially discussed by the scientific community and remains unclear, since a wide range of rates from 8–82% can be found in the literature [11,12,13,14]. Several factors for this phenomenon have been discussed. Using different definitions and criteria for diagnosing PCOS can lead to significant variations of the results. While most studies define RM as having three or more consecutive miscarriages, certain studies, especially the ones published before the implementation of the Rotterdam criteria as standardized definition for PCOS, primarily use the ovarian morphology alone for diagnosing PCOS [13,14,15,16], although the sole sonographic presence of polycystic ovaries does not describe the complete syndrome [11]. As demonstrated by Cocksedge et al., the prevalence is essentially lower using the Rotterdam criteria for diagnosing PCOS in RM population compared to studies using ultrasound only [11].

Certain features of PCOS are associated with an increased risk of RM, including hyperandrogenism, insulin resistance, hyperinsulinemia, obesity, elevated level of plasminogen-activator inhibitor (PAI)-1, and hyperhomocysteinemia [17]. Therefore, pathophysiological connections can be assumed and it is unclear whether PCOS would cause RM directly or whether the association is due to certain factors that are linked to both conditions. These uncertainties underline the need for further investigation.

Thus, the main goal of the present study was to evaluate the prevalence of polycystic ovarian morphology and PCOS within the RM population. Additionally, the data obtained in this study were included, together with previous reports on this topic, into a meta-analysis to provide a deeper understanding on how PCOS affects RM. As secondary study aims, we evaluated which recognized causes of RM were significantly associated with PCOS. We also investigated the outcome concerning pregnancy rates and baby-take-home rates compared to a matched control group.

## 2. Materials and Methods

### 2.1. Patient Population and Study Design of the Retrospective Cohort

In a retrospective cohort study, all women with RM who underwent evaluation at the Department of Gynecology and Obstetrics of the Medical University of Vienna, Vienna, Austria, from January 2003 to December 2018 were included (*n =* 452). RM was defined as three or more consecutive miscarriages before the 20th gestation week with the same partner [18]. The diagnostic evaluation for RM was performed according to the routine protocol of the Medical University of Vienna, previously published in Pils et al. [19]. Details are provided in the following section.

None of the PCOS women had laparoscopic ovarian drilling (LOD) before or after the study period. Twenty-eight women with PCOS and RM, who were negative for selected RM risk factors, as well as 28 age- and BMI-matched RM patients negative for selected factors but without PCOS conceived spontaneously after complete evaluation for RM and were included in an analysis on further pregnancy outcome. All of these patients received dydrogesterone 10 mg twice a day for luteal support during subsequent pregnancies until pregnancy week 12 + 0 [20].

The study was approved by the Institutional Review Board (IRB) of the Medical University of Vienna (IRB number 2151/2019). Data in this retrospective study were anonymized; thus, there was no need for informed consent according to the regulations of the IRB.

### 2.2. Parameters Analyzed

The main outcome parameter was PCOS, diagnosed according to the revised European Society of Human Reproduction and Embryology (ESHRE) and American Society for Reproductive Medicine (ASRM) criteria of 2004, which were based on the Rotterdam criteria [3]. According to these, at least two of the following three criteria must be fulfilled in order to establish a diagnosis: (i) oligo- and/or anovulation; (ii) clinical and/or biochemical signs of hyperandrogenism; and (iii) polycystic ovaries visualized with ultrasound. In addition, other possible causes for the symptoms, e.g., congenital adrenal hyperplasia, androgen-secreting tumors, hyperprolactinemia, thyroid disorders and Cushing’s syndrome, must be excluded [2,3]. In all women, non-classical adrenogenital syndrome was excluded by the use of the surrogate parameter 17-hydroxy progesterone <2 ng/mL.

Furthermore, we focused on the following parameters: age; primary or secondary RM; whether any of the following evidence-based or possible underlying causes of RM had been found during evaluation according to most recent guideline [18] (overt hypothyroidism, defined as Thyroid-stimulating hormone (TSH) > 5 µU/mL with free levothyroxine < 0.76 ng/dL, overt hyperthyroidism, defined as TSH < 0.4 µU/mL with free levothyroxine > 1.66 ng/dL, antiphospholipid-syndrome (APS), defined as the presence of at least one clinical criteria [≥1 venous or arterial thrombosis, or 1 or 2 unexplained miscarriages of morphologically normal fetuses after the 10th gestation week or ≥3 miscarriages before the 10th gestation week or ≥1 late miscarriage or preterm birth before the 34th gestation week because of placental insufficiency or preeclampsia] and one laboratory criteria [anti-cardiolipin IgM antibodies > 7 U/mL, anti-cardiolipin IgG antibodies > 10 U/mL or anti-beta-2-glycoprotein I IgM/IgG antibodies > 8 U/mL using repeated measures with a 12 week interval] [18], antithrombin deficiency, protein C and/or S deficiency, factor V Leiden-mutation, prothrombin deficiency, diabetes mellitus with HbA1c assessment, myomas/endometrial polyps/intrauterine adhesions/uterine malformations diagnosed/ruled out by vaginal sonography and diagnostic hysteroscopy, genetic factors, and presence of bacterial vaginosis or infection with ureaplasma urealyticum or mycoplasma hominis, whereas data on chronic endometritis were not available due to the historic cohort); whether RM was completely negative for selected risk factors when all of the above mentioned factors were absent; outcomes of further pregnancies (miscarriage or live birth after 23 + 0 week of gestation; for this parameter, multiple selections were possible if the patient had experienced more than one further pregnancy).

All blood samples were taken from a peripheral vein in a morning fasting state. All hormonal parameters were retrieved on the second to fifth cycle day. The examined serum parameters were determined in the central laboratory of the General Hospital of Vienna, Vienna, Austria using commercially available assays: For testosterone, the ELECSYS^®^ Testosterone II assay (Roche Diagnostics GmbH, Mannheim, Germany) was used; for androstenedione, the IMMULITE^®^ 2000 Androstenedione assay, (Siemens Healthcare Diagnostics Products Ltd., Llanberis, UK) was used; and for AMH, the DSL Active MIS/AMH assay (Beckman Coulter Inc., Brea, CA, USA), was used. All AMH values before 23 August 2013 were corrected using the following formula y = 2.01 × x (R2 = 0.98). This had been due to a complement interference problem of the Beckman AMH ELISA before August 2013, when a dilution protocol had been implemented [21].

### 2.3. Meta-Analysis

For the systematic literature review, we searched the Medline database (search date: 21 March 2020) to identify cohort studies, systematic reviews, and meta-analyses on recurrent miscarriage and polycystic ovary syndrome. The following search terms have been used to detect all relevant articles on this topic: (a) polycystic ovary syndrome AND recurrent miscarriage (*n* = 79); (b) “polycystic ovary syndrome” AND “recurrent miscarriage” (*n* = 39); (c) polycystic ovary syndrome AND recurrent pregnancy loss (*n* = 43); (d) “polycystic ovary syndrome” AND “recurrent pregnancy loss” (*n* = 33); and (e) prevalence polycystic ovary syndrome recurrent miscarriage (*n* = 24). Two authors assessed the eligibility of the studies, extracted data on PCOS prevalence in RM patients, and assessed the risk of bias (D.M. and J.O.). Missing information and additional trials were not sought from authors. Missing information and additional trials were not sought from authors.

Qualitative assessment of the studies included in the meta-analysis of studies with a sound GDM definition was also performed. Although not all items of the Newcastle-Ottawa Scale for cohort studies [22] were applicable for the meta-analysis, the items were used as far as possible. Concerning “selection”, we assessed whether the cohort was truly representative of the average recurrent miscarriage population. For the “comparability” of studies, we assessed whether age and BMI had been reported. For “outcome”, we evaluated whether the source for the retrospective data set had been specified. Studies fulfilling all criteria for were rated as having lowest risk for bias, studies fulfilling two of three items (“selection”, “comparability”, and “outcome”) were assessed as having a medium risk for bias and studies fulfilling one or no criterion were considered to have the highest risk for bias. Qualitative assessment of studies was performed by two researchers (D.M., J.O.).

### 2.4. Statistical Analysis

Variables are described by numbers (frequencies) and mean ± standard deviation. Statistical analysis was performed with SPSS 25.0 for Windows (SPSS Inc., Armonk, NY, USA ©1989–2019) using the unpaired t-test for numerical parameters and the Fisher’s exact test for categorical parameters. Odds ratios (OR) and their 95% confidence intervals (95% CI) are given. Differences were considered statistically significant if *p* < 0.05. The meta-analysis on the prevalence of PCOS in women with RM was performed as published previously [23]: the library “metafor” in the open source statistical package “*R*” (The *R* Project for Statistical Computing, Vienna, Austria) was used. The observed proportions were transformed using the Freeman–Tukey double arcsine transformation, which provides an effect measure with a favorable sampling distribution and stable variance. A meta-analysis model was fit to the transformed data using inverse variance weights and including a random effect to account for between-study heterogeneity. The random effects model was used due to the differences in observed PCOS prevalence in the included studies, in order not to underestimate the variability of data. A pooled estimate of the prevalence of PCOS and the corresponding 95% confidence interval were obtained by back-transforming the respective quantities to the original scale. Moreover, a leave-one-out sensitivity analysis and a funnel plot analysis to rule out publication bias were performed.

## 3. Results

### 3.1. Retrospective Cohort Study: Results of Complete Evaluation for Recurrent Miscarriage

PCOS was found in 43 (9.5%) of all women. Negative results for the selected risk factors for RM were present in 283 (62.6%). Mean age of the sample population was 33.8 ± 6.1 years, with a BMI of 24.9 ± 5.0 kg/m^2^. Primary RM was observed in 318 (70.4%) cases, secondary RM appeared in 134 (29.6%) cases. The majority of patients had three previous miscarriages (322, 71.2%); while 78 (17.3%) had four and 52 (70.4%) had five or more previous miscarriages. Basic patient characteristics and data on the underlying causes for RM are provided in Table 1. Only the presence of any thrombophilic defect was significantly associated with PCOS (PCOS: 20.9% versus no PCOS: 7.8%, *p =* 0.010; Appendix A).

### 3.2. Meta-Analysis on the Prevalence of Polycystic Ovarian Morphology and PCOS in RM

In total 93 articles were identified (excluding multiple mentions). For the meta-analysis concerning PCOS in RM patients, following articles were excluded step by step: Articles not published in English language (*n* = 6); reviews without original data (*n* = 24); studies and articles that did not report relevant finding for our analysis after full text review (*n* = 43); studies excluding patients with RM and/or PCOS (*n* = 3); animal studies (*n* = 1); case reports (*n* = 5); and articles using different definitions of RM and/or PCOS (*n* = 9).

As a result, only two studies were included in this meta-analysis [11,24], in addition to the present study. All the included studies defined RM as three or more consecutive miscarriages before the 20th week of gestation and used the revised Rotterdam criteria [3] for diagnosing PCOS. Notably, Matjila et al. [24] reported a PCOS prevalence of 22%, although PCOS was found in 148 of 592 women, which actually represented 25% of the study population. For the purpose of our meta-analysis, we chose to use the actual numbers provided by the authors (148/592 women with PCOS within the RM study population).

For the second meta-analysis evaluating the prevalence of polycystic ovarian morphology without additional use of the Rotterdam criteria for an exact PCOS definition, the following selection criteria for the initial 93 (excluding multiple mentions): Articles not published in English language (*n* = 6); reviews without original data (*n* = 24); studies and articles that did not report relevant finding for our analysis after full text review (*n* = 43); studies excluding patients with RM and/or PCOS (*n* = 3); animal studies (*n* = 1); case reports (*n* = 5); and articles using different definitions of RM and/or PCOS (*n* = 3).

Concerning the prevalence of PCOS defined by Rotterdam criteria, two studies in addition to our data set were eligible for the meta-analysis [11,24] and therefore included in the pooled models (Table 2). All three studies were of retrospective character and included a total of 1.344 women with RM. After correction for study heterogeneity, the estimated prevalence of PCOS was 14.3% (95% CI: 6.2–24.9) (Figure 1). For this analysis, two studies were assessed as having the lowest risk for bias [24] and this present study; one study having a medium risk for bias [11]. In the leave-one-out sensitivity analysis, the estimated prevalence of PCOS ranged from 9.7% to 16.9% (Appendix A).

Concerning the prevalence of polycystic ovarian morphology, eight/93 articles from the initial literature search had evaluated this outcome parameter in RM patients [13,14,15,16,25,26,27,28]. However, the following studies had to be excluded from the meta-analysis: the data of Clifford et al. [28] included in the more recent report of Rai et al. [14]; the study published by Okon et al. [27] due to the selected patient population of women with a cycle length of 25–35 days only and due to the fact that polycystic ovarian disease had been diagnosed either by ultrasound or by elevated early follicular phase LH concentration; the study conducted by Kousta et al. [26] had focused on the prevalence of polycystic ovaries among infertile women in general and not clearly in RM patients; Diejomaoh et al. [25] had used combined subgroups containing PCOS patients with and without infections (e.g., bacterial vaginosis) in their study and were therefore not considered eligible; last not least, one study had included women with a history of two or more consecutive miscarriages but without abnormal chromosomes in either partner, antiphospholipid antibodies or uterine abnormalities [12] and, thus, did not focus on an unselected RM population.

Accordingly, four studies remained which are listed in Table 2 and included 2.378 women with RM. After correction for study heterogeneity, the estimated prevalence of polycystic ovarian morphology was 50.8% (95% CI: 29.6–71.9). According to the above mentioned criteria, two studies were assessed as having the lowest risk for bias [15,16], one study as having a medium risk for bias [14] and one study fulfilling were considered to have the highest risk for bias [13]. A potential bias might be implied due to differences in the study design, namely the prospective and retrospective approaches used (Table 2). In the according funnel plot, three studies were not plotted near the average. In the leave-one-out sensitivity analysis, the estimated prevalence of polycystic ovarian morphology ranged from 40.5% to 56.0% (Appendix A).

### 3.3. Outcome of Women with PCOS and RM in the Retrospective Cohort Study

In a sub-analysis, 28 women with RM negative for selected risk factors and PCOS recruited from the initial study population were compared to 28 age- and BMI-matched women with RM but without PCOS. Details are shown in Table 3. Concerning basic patient, there were no differences between the groups. Women in the PCOS group revealed significantly higher LH, testosterone, and AMH levels (*p <* 0.05). The duration of follow-up was similar in both groups (PCOS: 18.6 ± 4.5 months versus non-PCOS: 18.3 ± 4.9 months; *p =* 0.799). Eight of the PCOS patients had undergone ovarian stimulation with clomiphene citrate to achieve pregnancy. The rate of further miscarriages was significantly higher in PCOS women than in controls (71.4% versus 53.6%, respectively; *p =* 0.031). Also, there was a difference of 18% between women with and without PCOS achieving a life birth (42.9% versus 60.7%, respectively; *p =* 0.285)., which did not reach statistical significance. The latter could be due to the small sample size of 28 vs. 28 patients.

## 4. Discussion

This study was carried out to gain further insight into the relationship between PCOS and RM. In a meta-analysis by Bozdag et al. [29] from 2016 containing 15 studies, the prevalence of PCOS in the general population, diagnosed using the revised Rotterdam criteria, was 10%. In our retrospective data set, 9.5% of all RM cases with three or more consecutive miscarriages were diagnosed with PCOS, whereas the meta-analysis showed an estimated pooled prevalence rate of 14.3% (95% CI: 6.2–24.9) (Figure 1). Thus, it seems likely that the rates of PCOS in RM patients are moderately higher than in the general population. This suggests that PCOS might play a role in RM. However, the prevalence in the studies included in our meta-analysis ranged from 9.5% to 25%.

Accordingly, these studies were not plotted near the average in the funnel plot analysis (Figure 2) some considerably clinically relevant variation of the pooled prevalence in the leave-one-out sensitivity analysis could be observed. The findings of Cocksedge et al. [11] and our data showed similar results (10.0% vs. 9.5%, respectively). Thus, the results of Matjila et al. [24] with a prevalence rate of 25.0% led to the relatively higher “overall” prevalence of PCOS in RM in the meta-analysis. Matjila et al. [24] assumed the relatively higher prevalence in their study to be the effect of a different application of the Rotterdam criteria, as Cocksedge et al. only used biochemical evidence for hyperandrogenism without considering clinical aspects of hyperandrogenism (e.g., by utilizing the Ferriman-Gallwey-Score for defining hirsutism, presence of acne or alopecia, etc.) which might have led to an underestimation of the real prevalence.

Another possible explanation might be the BMI, which was quite high in the study of Matjila et al. [24] compared to our data set (29.6 kg/m^2^ vs. 24.9 kg/m^2^, respectively). It seems noteworthy that several studies have confirmed a positive correlation between obesity and RM risk in spontaneous and recurrent miscarriage [30,31,32]. Possible pathophysiological factors include dysregulations of the hypothalamic-pituitary-ovarian axis, impaired oocyte quality, negative effects on preimplantation embryos, defects of the endometrial decidualization and a lower endometrial receptivity [32,33]. Unfortunately, the study by Cocksedge et al. [11] did not provide data on the BMI. In addition to these considerations, women with PCOS are more likely to develop metabolic conditions like insulin resistance or hyperinsulinemia [34] which are independent risk factors for miscarriage [10,35,36]. About 30% of women with PCOS are diagnosed with insulin resistance [37] and a prevalence of 27% insulin resistance was found in women suffering from RM [38]. Kazerooni et al. [39] demonstrated that the occurrence of hyperinsulinemia was higher in a group of women with RM and PCOS than in a group of women with RM and without PCOS. Tian et al. [40] stated that insulin resistance increased the risk of spontaneous abortion and the risk remained even higher with confounding factors, such as PCOS and obesity. In addition to the possible direct effects on RM, insulin and insulin-like growth factor stimulate androgen production in the ovary [38,41].

Nonetheless, the data of the meta-analysis showed no critically high prevalence of PCOS in RM women. This was in contrast to the findings of polycystic ovarian morphology only. In the second meta-analysis, 50.8% (95% CI: 29.6–71.9) of all RM patients presented with polycystic ovarian morphology. In this meta-analysis, also one specific study, namely Sagle et al. [13], was the major contributor for the high pooled prevalence (Figure 1). When focusing on the leave-one-out sensitivity analysis, the estimated prevalence noticeably dropped to 40.5% after exclusion of the mentioned study (Appendix A). In contrast, the meta-analysis by Bozdag et al. [29], which included 12 studies, demonstrated a prevalence for polycystic ovarian morphology of 28% in the general population. Compared to these results, our findings are considerably high, even if an actual prevalence of about 40% or 50% was assumed. Therefore, RM patients seem to be more likely to reveal polycystic ovarian morphology than women without RM. We find it hard to comment on this finding.

Despite discussions about the prevalence, the clinically most important question remains whether PCOS would influence pregnancy outcome in women with RM. To the best of our knowledge, reproductive outcomes of these women have been reported only once, yet with the focus on patients with polycystic ovarian morphology instead of Rotterdam-defined PCOS. In this study, Rai et al. [14] found no significant differences regarding overall live birth rates, mean gestational age at the time of delivery, and mean birth weight compared to women with RM but without PCOM. Therefore, our report is the first on Rotterdam-defined PCOS and miscarriage/live birth rates after RM. It is not surprising that women with PCOS revealed significantly higher LH, testosterone, and AMH levels (*p* < 0.05) than the control group without PCOS (Table 3). Notably, women with PCOS were significantly more likely to experience a further miscarriage (71.4% versus 53.6%; *p* = 0.031). These findings are consistent with previous reports which showed that women with PCOS had a higher risk for first trimester miscarriages [42]. Importantly, none of the evaluated women in our study received treatment for PCOS prior to or at the time of investigation, except for luteal support with dydrogesterone during pregnancy. However, in PCOS women the live birth rate was about 18% lower than in women with RM negative for selected risk factors. However, this finding did not reach statistical significance. One could argue that having chosen a control group of women with RM negative for selected risk factors and the small sample size in this analysis (28 vs. 28 women) should be considered study limitations, the results seem promising for PCOS patients from a clinical point of view. As mentioned above, these outcome data only refer to RM women without selected risk factors. We considered it valuable to select only these women in order to assess the influence of PCOS alone. Notably, concerning the suggested underlying causes for RM in our data set, PCOS was only significantly associated with thrombophilic defects (Appendix A). In literature, numerous variations of coagulation disorders have been the subject of investigation, but only antiphospholipid syndrome showed a strong correlation with RM [43], while results on the role of other thrombophilic defects, namely Factor V Leiden, prothrombin deficiency, antithrombin III deficiency, protein C deficiency and protein S deficiency, were conflicting [44,45]. In our study, thrombophilic defects did not show a significant association until combining all thrombophilic disorders in one category. One might argue that hyperhomocysteinemia should be included in the thrombophilia assessment. This was not done in the patients of our data set, since there is no recent recommendation on this in Austria. We consider this circumstance a minor study limitation. Noteworthy, in our data set, PCOS was significantly associated with RM without selected risk factors (Appendix A). Although one cannot rule out that PCOS might be of importance in RM due to its association with other underlying causes for RM, our data lend support to the hypothesis that it is of influence itself.

Last but not least, several study limitations have to be considered: the retrospective design of our cohort study; the small sample size of the case-control sub-analysis on pregnancy outcome; moreover, some women with PCOS had undergone stimulation with clomiphene citrate to achieve pregnancy after evaluation for RM and, hypothetically, there might be a difference between women who manage to conceive naturally and those in need for ovarian stimulation and ovulation induction; during the study period, chronic endometritis was not evaluated which is due to the historical population. This seems worth mentioning, since the effective pharmacologic treatment of chronic endometritis seems to improve pregnancy and life birth rates of patients in patients with RM negative for selected risk factors [46]; concerning the meta-analysis, the small number of studies included is a limitation as well as the moderate to high risk of bias in some of these studies, which might be reflected by the wide ranges for the pooled PCOS and polycystic ovarian morphology prevalence in the leave-one-out sensitivity analyses. Another minor limitation is the short follow-up period for pregnancy outcome of 18.6 months; however we think this approach is superior to estimate the outcome than looking at the outcome of the subsequent pregnancy alone. The prevalence of bacterial vaginosis and parental chromosomal abnormalities were surprisingly low in our patient population. We find it hard to comment on this phenomenon. However, we cannot rule out a selection bias of women who were referred to our department. We consider the fact that only women with ≥3 previous miscarriages were defined as having RM a strength of our analysis.

## 5. Conclusions

The prevalence of PCOS seems slightly increased in women with RM, PCOM on the other hand show a rather high prevalence compared to the general population. PCOS women who suffered from RM were at significantly higher risk for further miscarriage and revealed decreased chances of having a life birth by about 18%, which did not reach statistical significance. Thus, PCOS might play a moderate role in RM. However, future studies are warranted, especially concerning pregnancy outcomes after RM in women with PCOS and the influence of PCOS-specific treatment on these issues.

## Figures and Tables

**Figure 1 jcm-09-02700-f001:**
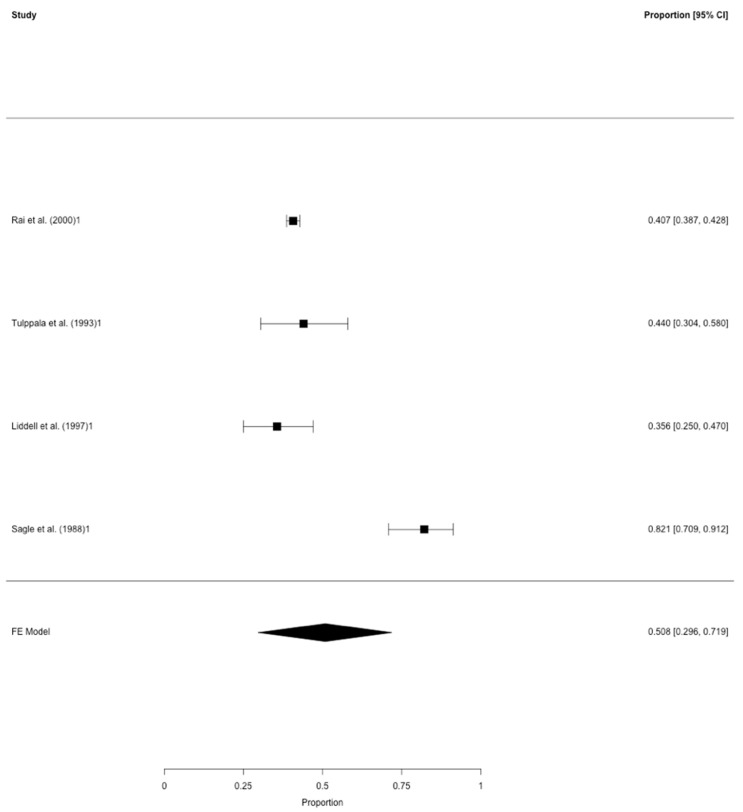
The prevalence of polycystic ovarian morphology in RM. Meta-analysis of four studies on polycystic ovarian morphology in patients with RM (≥3 consecutive miscarriages before gestation week 20). Provided are the proportions including the 95% confidence intervals.

**Figure 2 jcm-09-02700-f002:**
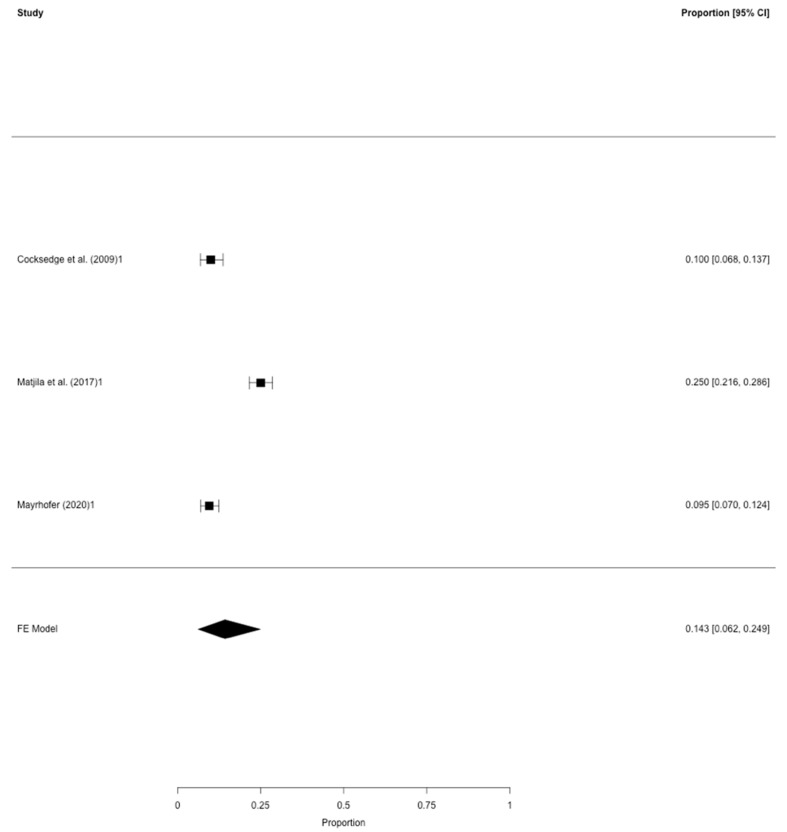
The prevalence of PCOS in RM. Meta-analysis of all three studies on PCOS (defined by Rotterdam criteria) in patients with RM (≥3 consecutive miscarriages before gestation week 20). Provided are the proportions including the 95% confidence intervals.

**Table 1 jcm-09-02700-t001:** Basic patient characteristics and results of complete evaluation of RM in the retrospective cohort (*n =* 452).

**Age (years) ***		33.8 ± 6.1
**BMI (kg/m^2^) ***		24.9 ± 5.0
Number of previous miscarriages ^#^	3	322 (71.2)
	4	78 (17.3)
	≥5	52 (11.5)
Type of recurrent miscarriage ^#^	Primary	318 (70.4)
	Secondary	134 (29.6)
PCOS ^#^		43 (9.5)
*Underlying causes for RM*^+^:		
Recurrent miscarriage negative for selected risk factors ^#^		181 (40.0)
Recurrent miscarriage negative for selected risk factors (includes women with PCOS only)		209 (46.2)
BMI >25 kg/m^2^		156 (34.5)
Hypothyroidism ^#^		16 (13.5)
Hyperthyroidism ^#^		11 (2.4)
Antiphospholipid syndrome ^#^		17 (3.8)
Antithrombin deficiency ^#^		7 (1.5)
Factor V Leiden-mutation ^#^		16 (3.5)
Prothrombin deficiency ^#^		5 (1.1)
Protein C deficiency ^#^		7 (1.5)
Protein S deficiency ^#^		9 (2.0)
Bacterial vaginosis (including infection with ureaplasm and mycoplasma hominis)		23 (5.1)
Uterine malformation (septate or bicornuate uterus) ^#^		25 (5.5)
Uterine abnormalities (endometrial polyp, myoma, intrauterine synechia)		43 (9.5)
Parental chromosomal abnormalities		1 (0.2)

Data are provided by * mean ± standard deviation or ^#^ number (frequency). ^+^ Multiple citations possible.

**Table 2 jcm-09-02700-t002:** Characteristics and results of the studies on the prevalence of polycystic ovarian morphology and PCOS in RM included in the meta-analyses.

First Author	Year	Study Design	PCO Morphology Definition	Total Number of Women with RM	Number of Women with PCOM */PCOS	Rate of Women with PCOM */PCOS ^#^
**Studies on polycystic ovarian morphology (PCOM)**						
Sagle et al. [13]	1988	retrospective	≥10 cysts of 2–8 mm	56	46	82.1
Tulppala et al. [15]	1993	prospective	≥10 cysts of ≥2 mm	50	22	44.0
Liddell et al. [16]	1997	prospective	≥10 cysts of ≥2 mm	73	26	35.6
Rai et al. [14]	2000	retrospective	ovarian volume >9 mL, ≥10 cysts of 2–8 mm	2199	895	40.7
**Studies on polycystic ovarian syndrome (PCOS)**						
Cocksedge et al. [11]	2009	retrospective	-	300	30	10.0
Matjila et al. [24]	2017	retrospective	-	592	148	25.0
Mayrhofer et al.	2020	retrospective	-	452	43	9.5

Abbreviations used: * PCOM, polycystic ovarian morphology; ^#^ PCOS, polycystic ovarian syndrome.

**Table 3 jcm-09-02700-t003:** Basic patient characteristics and further pregnancy outcome in women with RM negative for selected risk factors with and without PCOS (retrospective cohort).

		RM Negative for Selected Risk Factors with PCOS (*n =* 28)	RM Negative for Selected Risk Factors without PCOS (*n =* 28)	*p*
**Age (years) ***		29.9 ± 4.8	29.9 ± 4.5	0.989
BMI (kg/m^2^) *		25.6 ± 5.1	25.1 ± 4.6	0.669
Number of previous miscarriages ^#^	3	20 (71.4)	20 (71.4)	1.000
	4	5 (17.9)	6 (21.4)	
	≥5	3 (10.7)	2 (7.1)	
Type of recurrent miscarriage ^#^	Primary	20 (71.4)	17 (60.7)	0.573
	Secondary	8 (28.6)	11 (39.3)	
FSH (mIU/mL) *		5.3 ± 2.1	5.6 ± 2.8	0.656
LH (mIU/mL) *		8.8. ± 4.1	5.9 ± 3.8	*0.009 ^+^*
LH:FSH ratio		1.9 ± 1.1	1.3 ± 1.3	0.125
Total testosterone (ng/mL) *		0.61 ± 0.12	0.28 ± 0.10	*<0.001 ^+^*
AMH (ng/mL) *		5.59 ± 2.62	2.12 ± 1.89	*<0.001 ^+^*
Follow-up period (months) *		18.6 ± 4.5	18.3 ± 4.9	0.799
One or more further miscarriages ^#^		20 (71.4)	15 (53.6)	*0.031 ^+^*
One or more live births ^#^		12 (42.9)	17 (60.7)	0.285

Data are provided by * mean ± standard deviation or ^#^ number (frequency). ^+^ Significant *p*-values are provided in italic font. Abbreviations used: RM, recurrent miscarriage; PCOS, polycystic ovary syndrome; BMI, body mass index; FSH, follicle-stimulation hormone; LH, luteinizing hormone; AMH, anti-Mullerian hormone.

## Data Availability

The datasets generated and/or analyzed during the current study are not publicly available, since the dataset will be used for other retrospective analyses. The data are available from the corresponding author upon reasonable request.

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
