# Peer review of "The Prevalence and Impact of Polycystic Ovary Syndrome in Recurrent Miscarriage: A Retrospective Cohort Study and Meta-Analysis"

_jcm, 2020, doi:10.3390/jcm9092700_

Round 1

Reviewer 1 Report

This article on the prevalence and impact of PCOS on MRs is interesting and well documented. The idea is to do two meta-analysis, one based on the definition of PCOS according to the Rotterdam criteria (three studies, including the author's retrospective study) and the other on multifolmolcular ultrasound aspects (PCOS ultrasound aspect) (4 studies).

The article therefore begins with a retrospective study (452 women) (attention the author's name is given in the meta-analysis - replace by our study), which finds 9.3% of PCOS in case of RM, increasing to 14.3% in the first meta-analysis.

One wonders why homocysteine was not dosed in the thrombophilia balance.

Page 7: The last paragraph must be located in the 3rd paragraph to give more clarity to the subject.

Figure 3 adds nothing to Figures 1 and 2: so Figure 3 to remove.

In the discussion, it is alluded to the role of BMI, but never insulin resistance: to develop.

Author Response

Dear Reviewer,

We thank you for your comments and recommendations regarding our manuscript “The prevalence and impact of polycystic ovary syndrome in recurrent miscarriage: a retrospective cohort study and meta-analysis”. We took care in revising our work according to your suggestions and hope that the revisions made will make our manuscript acceptable for publication in the Journal of Clinical Medicine.

Please find the Point-by-Point answer letter below. The manuscript was proof-read by an English native speaker.

We thank you for your interest in our manuscript,

Respectfully yours,

Johannes Ott

One wonders why homocysteine was not dosed in the thrombophilia balance.

Reply: We thank the reviewer for this question. This is due to the fact that recent guidelines for RM in Germany and Austria did not address hyperhomocysteinemia as a relevant factor. In the revised manuscript, we address this circumstance as a minor study limitation: “One might argue that hyperhomocysteinemia should be included in the thrombophilia assessment. This was not done in the patients of our data set, since there is no recent recommendation on this in Austria. We consider this circumstance a minor study limitation.”

Page 7: The last paragraph must be located in the 3rd paragraph to give more clarity to the subject.

Reply: Revised as recommended.

Figure 3 adds nothing to Figures 1 and 2: so Figure 3 to remove.

Reply: Removed.

In the discussion, it is alluded to the role of BMI, but never insulin resistance: to develop.

Reply: Thank you for your valuable comment. We added the following paragraph to the Discussion Section: “In addition to these considerations, women with PCOS are more likely to develop metabolic conditions like insulin resistance or hyperinsulinemia [34] which are independent risk factors for miscarriage [10, 35, 36]. About 30% of women with PCOS are diagnosed with insulin resistance [37] and a prevalence of 27% insulin resistance was found in women suffering from RM [38]. Kazerooni et al. [39] demonstrated that the occurrence of hyperinsulinemia was higher in a group of women with RM and PCOS than in a group of women with RM and without PCOS. Tian et al. [40] stated that insulin resistance increased the risk of spontaneous abortion and the risk remained even higher with confounding factors, such as PCOS and obesity. In addition to the possible direct effects on RM, insulin and insulin-like growth factor stimulate androgen production in the ovary [38, 41].”

Reviewer 2 Report

The retrospective cohort study and meta-analysis by Mayrhofer et al. are important contributions to our knowledge about the role of PCOS in recurrent miscarriage RM). The prevalence of PCOS in RM and its impact on pregnancy outcome are very disputed.

Although the study has several weaknesses, the strengths are dominant.

It is good that in the authors’ own cohort, RM is defined as at least 3 consecutive miscarriages and they define PCOS according to the Rotterdam criteria.

It is a valid and useful information that they find a 9.5% PCOS rate in their cohort of 452 RM patients. This prevalence is in accordance with that found in my clinic-

The statistics seems to be OK. The follow-up period for pregnancy outcome of 18.6 months may be short but is a much better way to estimate outcome that to look at outcome in the next pregnancy.

Some of the weaknesses that should be corrected or commented are:

The authors like many other make the error that RM patients positive for some risk factors for RM are called “unexplained” and they refer to the 50% rate of “unexplained” RM quoted in many papers, which have never been documented. Indeed, as realized in international RM guidelines such as the ESHRE Guideline on RM only the full-blown antiphospholipid syndrome and clinical hypo- and hyperthyroidism (and repeated embryonal aneuploidy that are rarely documented) are recognized as documented causal factors for RM and should be routinely investigated. I will recommend that the authors in the abstract, text and tables instead of using the expression “unexplained” and “explained” RM use the expression “negative for selected risk factors” and “positive for a least one risk factor”.

Some of the risk facts that were investigated are poorly described or have surprisingly low prevalences:

How do the authors define overt hypo- and hyperthyroidism?

How do they define antiphospholipid syndrome? which cutoff values? Repeated measurement with 12 weeks intervals?

The prevalence of bacterial vaginosis is surprisingly low. In my country almost 50% of patients are positive for ureaplasma urealyticum in vaginal swaps so therefore it cannot be considered a risk factor but belongs to the normal vaginal flora.

The low prevalence (0.2%) of parental chromosomal abnormalities is also surprising; in practically all other studies a prevalence of 5% has been reported. Some kind of genetic selection of the patients must have been undertaken before referral to the authors’ clinic. Please comment.

It is surprising that the authors could not collect at larger control group than 28 for evaluation of pregnancy outcome in PCOS positive and negative women. The authors’ conclusion that subsequent birth rate is not affected by PCOS state is not valid, there is a 18% difference in birth rate between the two groups, but the study is clearly statistically underpowered with regard to subsequent live birth rate. This must be changed!

Figures 1 and 2 are difficult to understand, please provide more information in the figures and legends

Author Response

Dear Reviewer,

We thank you for your comments and recommendations regarding our manuscript “The prevalence and impact of polycystic ovary syndrome in recurrent miscarriage: a retrospective cohort study and meta-analysis”. We took care in revising our work according to your suggestions and hope that the revisions made will make our manuscript acceptable for publication in the Journal of Clinical Medicine.

Please find the Point-by-Point answer letter below. The manuscript was proof-read by an English native speaker.

We thank you for your interest in our manuscript,

Respectfully yours,

Johannes Ott

The retrospective cohort study and meta-analysis by Mayrhofer et al. are important contributions to our knowledge about the role of PCOS in recurrent miscarriage RM). The prevalence of PCOS in RM and its impact on pregnancy outcome are very disputed. Although the study has several weaknesses, the strengths are dominant.

Reply: We thank the reviewer for the positive overall statement about our work.

It is good that in the authors’ own cohort, RM is defined as at least 3 consecutive miscarriages and they define PCOS according to the Rotterdam criteria.

It is a valid and useful information that they find a 9.5% PCOS rate in their cohort of 452 RM patients. This prevalence is in accordance with that found in my clinic-

The statistics seems to be OK. The follow-up period for pregnancy outcome of 18.6 months may be short but is a much better way to estimate outcome that to look at outcome in the next pregnancy.

Reply: We agree and added a cooresponding statement to the Discussion Section. We took the liberty to use the reviewer’s wording and hope that this is okay for the reviewer: “Another minor limitation is the short follow-up period for pregnancy outcome of 18.6 months; however we think this approach is superior to estimate the outcome than looking at the outcome of the subsequent pregnancy alone.”

Some of the weaknesses that should be corrected or commented are:

The authors like many other make the error that RM patients positive for some risk factors for RM are called “unexplained” and they refer to the 50% rate of “unexplained” RM quoted in many papers, which have never been documented. Indeed, as realized in international RM guidelines such as the ESHRE Guideline on RM only the full-blown antiphospholipid syndrome and clinical hypo- and hyperthyroidism (and repeated embryonal aneuploidy that are rarely documented) are recognized as documented causal factors for RM and should be routinely investigated. I will recommend that the authors in the abstract, text and tables instead of using the expression “unexplained” and “explained” RM use the expression “negative for selected risk factors” and “positive for a least one risk factor”.

Reply: Corrected throughout the manuscript.

Some of the risk facts that were investigated are poorly described or have surprisingly low prevalences:

How do the authors define overt hypo- and hyperthyroidism?

Reply: Thank you for this suggestion. We added the following information to the manuscript: “[…] defined as TSH> 5µU/mL with free levothyroxine< 0.76ng/dL, overt hyperthyroidism, defined as TSH< 0.4µU/mL with free levothyroxine> 1.66ng/dL […]”

How do they define antiphospholipid syndrome? which cutoff values? Repeated measurement with 12 weeks intervals?

Reply: We clarified this issue as follows:

Methods: “[…] antiphospholipid-syndrome (APS), defined as the presence of at least one clinical criteria [≥1 venous or arterial thrombosis, or 1 or 2 unexplained miscarriages of morphologically normal fetuses after the 10th gestation week or ≥ 3 miscarriages before the 10th gestation week or ≥1 late miscarriage or preterm birth before the 34th gestation week because of placental insufficiency or preeclampsia] and one laboratory criteria [anti-cardiolipin IgM antibodies> 7U/mL, anti-cardiolipin IgG antibodies> 10U/mL or anti-beta-2-glycoprotein I IgM/IgG antibodies> 8U/mL using repeated measures with a 12 week interval] [18] […]”

The prevalence of bacterial vaginosis is surprisingly low. In my country almost 50% of patients are positive for ureaplasma urealyticum in vaginal swaps so therefore it cannot be considered a risk factor but belongs to the normal vaginal flora.

The low prevalence (0.2%) of parental chromosomal abnormalities is also surprising; in practically all other studies a prevalence of 5% has been reported. Some kind of genetic selection of the patients must have been undertaken before referral to the authors’ clinic. Please comment.

Reply: We comment on this in the Discussion Section, paragraph on study limitations: “The prevalence of bacterial vaginosis and parental chromosomal abnormalities were surprisingly low in our patient population. We find it hard to comment on this phenomenon. However, we cannot rule out a selection bias of women who were referred to our department.”

It is surprising that the authors could not collect at larger control group than 28 for evaluation of pregnancy outcome in PCOS positive and negative women. The authors’ conclusion that subsequent birth rate is not affected by PCOS state is not valid, there is a 18% difference in birth rate between the two groups, but the study is clearly statistically underpowered with regard to subsequent live birth rate. This must be changed!

Reply: We thank the reviewer for raising this important issue. We changed this conclusion throughout the manuscript (Abstract, Discussion, and Conclusion Sections):

Abstract: “Women with PCOS suffering from RM showed a significantly higher risk for further miscarriage and decreased chances of having a life birth of about 18% which did not reach statistical significance.”

Discussion: “However, in PCOS women the live birth rate was about 18% lower than in women with RM negative for selected risk factors. However, this finding did not reach statistical significance. One could argue that having chosen a control group of women with RM negative for selected risk factors and the small sample size in this analysis (28 vs. 28 women) should be considered study limitations, the results seem promising for PCOS patients from a clinical point of view.”

Final conclusion: “PCOS women who suffered from RM were at significantly higher risk for further miscarriage and revealed decreased chances of having a life birth by about 18%, which did not reach statistical significance.”

Figures 1 and 2 are difficult to understand, please provide more information in the figures and legends

Reply: We extended the Figure legends as follows including a new main title for each Figure:

“Figure 1. The prevalence of polycystic ovarian morphology in RM. Meta-analysis of four studies on polycystic ovarian morphology in patients with RM (≥3 consecutive miscarriages before gestation week 20). Provided are the proportions including the 95% confidence intervals.”

“Figure 2. The prevalence of PCOS in RM. Meta-analysis of all three studies on PCOS (defined by Rotterdam criteria) in patients with RM (≥3 consecutive miscarriages before gestation week 20). Provided are the proportions including the 95% confidence intervals.”

Reviewer 2

The retrospective cohort study and meta-analysis by Mayrhofer et al. are important contributions to our knowledge about the role of PCOS in recurrent miscarriage RM). The prevalence of PCOS in RM and its impact on pregnancy outcome are very disputed. Although the study has several weaknesses, the strengths are dominant.

Reply: We thank the reviewer for the positive overall statement about our work.

It is good that in the authors’ own cohort, RM is defined as at least 3 consecutive miscarriages and they define PCOS according to the Rotterdam criteria.

It is a valid and useful information that they find a 9.5% PCOS rate in their cohort of 452 RM patients. This prevalence is in accordance with that found in my clinic-

The statistics seems to be OK. The follow-up period for pregnancy outcome of 18.6 months may be short but is a much better way to estimate outcome that to look at outcome in the next pregnancy.

Reply: We agree and added a cooresponding statement to the Discussion Section. We took the liberty to use the reviewer’s wording and hope that this is okay for the reviewer: “Another minor limitation is the short follow-up period for pregnancy outcome of 18.6 months; however we think this approach is superior to estimate the outcome than looking at the outcome of the subsequent pregnancy alone.”

Some of the weaknesses that should be corrected or commented are:

The authors like many other make the error that RM patients positive for some risk factors for RM are called “unexplained” and they refer to the 50% rate of “unexplained” RM quoted in many papers, which have never been documented. Indeed, as realized in international RM guidelines such as the ESHRE Guideline on RM only the full-blown antiphospholipid syndrome and clinical hypo- and hyperthyroidism (and repeated embryonal aneuploidy that are rarely documented) are recognized as documented causal factors for RM and should be routinely investigated. I will recommend that the authors in the abstract, text and tables instead of using the expression “unexplained” and “explained” RM use the expression “negative for selected risk factors” and “positive for a least one risk factor”.

Reply: Corrected throughout the manuscript.

Some of the risk facts that were investigated are poorly described or have surprisingly low prevalences:

How do the authors define overt hypo- and hyperthyroidism?

Reply: Thank you for this suggestion. We added the following information to the manuscript: “[…] defined as TSH> 5µU/mL with free levothyroxine< 0.76ng/dL, overt hyperthyroidism, defined as TSH< 0.4µU/mL with free levothyroxine> 1.66ng/dL […]”

How do they define antiphospholipid syndrome? which cutoff values? Repeated measurement with 12 weeks intervals?

Reply: We clarified this issue as follows:

Methods: “[…] antiphospholipid-syndrome (APS), defined as the presence of at least one clinical criteria [≥1 venous or arterial thrombosis, or 1 or 2 unexplained miscarriages of morphologically normal fetuses after the 10th gestation week or ≥ 3 miscarriages before the 10th gestation week or ≥1 late miscarriage or preterm birth before the 34th gestation week because of placental insufficiency or preeclampsia] and one laboratory criteria [anti-cardiolipin IgM antibodies> 7U/mL, anti-cardiolipin IgG antibodies> 10U/mL or anti-beta-2-glycoprotein I IgM/IgG antibodies> 8U/mL using repeated measures with a 12 week interval] [18] […]”

The prevalence of bacterial vaginosis is surprisingly low. In my country almost 50% of patients are positive for ureaplasma urealyticum in vaginal swaps so therefore it cannot be considered a risk factor but belongs to the normal vaginal flora.

The low prevalence (0.2%) of parental chromosomal abnormalities is also surprising; in practically all other studies a prevalence of 5% has been reported. Some kind of genetic selection of the patients must have been undertaken before referral to the authors’ clinic. Please comment.

Reply: We comment on this in the Discussion Section, paragraph on study limitations: “The prevalence of bacterial vaginosis and parental chromosomal abnormalities were surprisingly low in our patient population. We find it hard to comment on this phenomenon. However, we cannot rule out a selection bias of women who were referred to our department.”

It is surprising that the authors could not collect at larger control group than 28 for evaluation of pregnancy outcome in PCOS positive and negative women. The authors’ conclusion that subsequent birth rate is not affected by PCOS state is not valid, there is a 18% difference in birth rate between the two groups, but the study is clearly statistically underpowered with regard to subsequent live birth rate. This must be changed!

Reply: We thank the reviewer for raising this important issue. We changed this conclusion throughout the manuscript (Abstract, Discussion, and Conclusion Sections):

Abstract: “Women with PCOS suffering from RM showed a significantly higher risk for further miscarriage and decreased chances of having a life birth of about 18% which did not reach statistical significance.”

Discussion: “However, in PCOS women the live birth rate was about 18% lower than in women with RM negative for selected risk factors. However, this finding did not reach statistical significance. One could argue that having chosen a control group of women with RM negative for selected risk factors and the small sample size in this analysis (28 vs. 28 women) should be considered study limitations, the results seem promising for PCOS patients from a clinical point of view.”

Final conclusion: “PCOS women who suffered from RM were at significantly higher risk for further miscarriage and revealed decreased chances of having a life birth by about 18%, which did not reach statistical significance.”

Figures 1 and 2 are difficult to understand, please provide more information in the figures and legends

Reply: We extended the Figure legends as follows including a new main title for each Figure:

“Figure 1. The prevalence of polycystic ovarian morphology in RM. Meta-analysis of four studies on polycystic ovarian morphology in patients with RM (≥3 consecutive miscarriages before gestation week 20). Provided are the proportions including the 95% confidence intervals.”

“Figure 2. The prevalence of PCOS in RM. Meta-analysis of all three studies on PCOS (defined by Rotterdam criteria) in patients with RM (≥3 consecutive miscarriages before gestation week 20). Provided are the proportions including the 95% confidence intervals.”
